# Clinical Profile, Treatment and Predictors during the First COVID-19 Wave: A Population-Based Registry Analysis from Castile and Leon Hospitals

**DOI:** 10.3390/ijerph17249360

**Published:** 2020-12-14

**Authors:** Eduardo Gutiérrez-Abejón, Eduardo Tamayo, Débora Martín-García, F. Javier Álvarez, Francisco Herrera-Gómez

**Affiliations:** 1Pharmacological Big Data Laboratory, Faculty of Medicine, University of Valladolid, 47005 Valladolid, Spain; alvarez@med.uva.es (F.J.Á.); fherrerag@saludcastillayleon.es (F.H.-G.); 2Technical Direction of Pharmaceutical Assistance, Gerencia Regional de Salud de Castilla y León, 47007 Valladolid, Spain; 3BioCritic. Group for Biomedical Research in Critical Care Medicine, 47005 Valladolid, Spain; tamayo@med.uva.es; 4Department of Anaesthesiology, Hospital Clínico Universitario de Valladolid, 47003 Valladolid, Spain; 5Department of Surgery, Faculty of Medicine, University of Valladolid, 47005 Valladolid, Spain; 6Department of Nephrology, Hospital Clínico Universitario de Valladolid, 47003 Valladolid, Spain; deboramarg@yahoo.es; 7CEIm, Hospital Clínico Universitario de Valladolid, 47003 Valladolid, Spain; 8Department of Nephrology, Hospital Virgen de la Concha, 49022 Zamora, Spain

**Keywords:** SARS-CoV-2, COVID-19, clinical characteristics, treatment, mortality, severe acute respiratory syndrome, acute kidney injury

## Abstract

The first wave of the COVID-19 pandemic collapsed the hospitals in Castile and Leon (Spain). An analysis of the clinical characteristics, drug therapies and principal outcome predictors in the COVID-19 hospitalized patients from 1 March to 31 May 2020 is presented through a population-based registry study. Hospital stay variables, ventilation mode data and clinical outcomes were observed. In Castile and Leon hospitals, 7307 COVID-19 patients were admitted, with 57.05% being male and a median of 76 years. The mortality rate was 24.43%, with a high incidence of severe acute respiratory syndrome (SARS) (14.03%) and acute kidney injury (AKI) (10.87%). The most used medicines were antibiotics (90.83%), antimalarials (42.63%), steroids (44.37%) and antivirals, such as lopinavir/ritonavir (42.63%). The use of tocilizumab (9.37%) and anti-SIRS (systemic inflammatory response syndrome) medicines (7.34%) were remarkable. Fundamentally, death occurred more likely over 65 years of age (OR: 9.05). In addition, the need for ventilation was associated with a higher probability of death (OR: 3.59), SARS (OR: 5.14) and AKI (OR: 2.31). The drug-use pattern had been modified throughout the COVID-19 first wave. Multiple factors, such as age, gender and the need for mechanical ventilation, were related to the worst evolution prognosis of the disease.

## 1. Introduction

The outbreak of a novel coronavirus in December 2019, which originated in Wuhan (China) and later spread to the rest of the world, represents a great threat to global health [1,2]. It is an RNA betacoronavirus, denominated as severe acute respiratory syndrome coronavirus 2 (SARS-CoV-2), belonging to the same family as SARS-CoV and the Middle East respiratory syndrome coronavirus (MERS-CoV) [3,4,5].

In February 2020, the World Health Organization (WHO) named the disease as COVID-19 (coronavirus disease 2019) [6], and on 11 March 2020 a pandemic was declared due to the increase in the number of persons infected outside China [7,8].

The symptoms of COVID-19 range from asymptomatic infection to severe acute respiratory syndrome (SARS), sepsis, and multi-organ failure, with high mortality rates of around 2–5% [3,4,9,10]. The clinical picture includes fever, headache, dry cough, dyspnea, fatigue and conjunctivitis. Lymphopenia, elevated D-dimer and other analytical signs of systemic inflammatory response syndrome (SIRS) has been observed [4,9]. In severe cases, individuals present severe bilateral interstitial pneumonia, commonly requiring admission in intensive care units (ICU) and mechanical ventilation [3,4,11,12,13]. Furthermore, SARS-CoV-2, like other coronaviruses, not only affects the respiratory tract but also the digestive tract, liver and heart [14].

In the first COVID-19 wave, between March and May 2020, pharmacological treatment focused on the relief of symptoms and the protection of the multiple organs affected, since there was not a vaccine or specific antiviral treatment [15]. Thus, treatments used in the diseases caused by SARS-CoV and MERS-CoV, and other experimental drugs, were tried [4].

Currently, there are no formal or specific treatments. The only antiviral drug authorized by the Food and Drug Administration (FDA) and the European Medicines Agency (EMA) for the treatment of SARS-CoV-2 is remdesivir, an adenosine nucleotide analog that interferes with RNA polymerization [16].

According to the Spanish national and regional health authorities [17,18], medicines used in the treatment of COVID-19 in the first COVID-19 wave were the following: (1) antibiotics, such as azithromycin or ceftriaxone; (2) antimalarials, such as hydroxychloroquine or chloroquine; (3) steroids, such as methylprednisolone or prednisone; (4) antivirals, such as lopinavir/ritonavir or remdesivir; and (5) anti-SIRS drugs, such as interleukin 1 (IL-1) inhibitors, such as anakinra, interleukin 6 (IL-6) inhibitors, such as tocilizumab, sarilumab or siltuximab, selective inhibitors of Janus-associated kinases (JAK), such as ruxolitinib or baricitinib, and several types of interferon (IFN), such as IFN β-1b and IFN α-2b [17,18,19,20,21,22,23,24,25,26,27,28,29,30]. Importantly, in our country the stocks for various anti-SIRS drugs became spent rapidly, and decisions to use them in patients with a predictable better prognosis were necessary. In these cases, most of these medicines were used off the label.

Focusing on the Castile and Leon public healthcare systems, the network of hospitals welcoming COVID-19 patients comprises 14 hospitals, of which 3 are regional hospitals, 6 are midsize or general hospitals and 5 are first-level referral hospitals (Appendix A). The total capacity of the hospital network was 7141 beds to cover a population of 2,323,770 inhabitants. Each of these 14 hospitals established its treatment strategy based on the Spanish national guidelines described previously [17,18].

The aim of this study was to assess patterns in the use of medications to treat in-hospital COVID-19 patients, and how these patterns were related to mortality, as well as the occurrence of SARS, acute kidney injury (AKI), cardiomyopathy and shock. With this aim, we have accessed clinical records of all in-hospital COVID-19 patients treated during the first wave of the disease, from 1 March to 31 May 2020 in Castile and Leon, Spain.

## 2. Results

### 2.1. Clinical Findings

In this study, 7307 hospitalized COVID-19 patients were included, of which 57.05% were men and 42.95% women (*p* = 0.001). The median (interquartile range: IQR) age was 76 (63–86) years, being higher in women than in men (*p* = 0.001), and 72.77% of patients were 65 years or older (Table 1).

Invasive mechanical ventilation (IMV) was required by 3.5% of COVID-19 patients and 1.63% required noninvasive positive pressure ventilation (NIPPV), while 2.52% received only oxygenation, with overall ventilation requirements higher in men than in women (Table 1).

The median hospitalization period was 9 (5–15) days, significantly longer in men than in women (*p* = 0.001), but no differences were found between women and men in terms of length of stay in the ICU, with a median of 15 (7–30) days (Table 1).

Highest incidence was observed in SARS (14.03%) and AKI (10.87%), and the mortality rate was 24.43%, being higher in men than in women (*p* = 0.001) (Table 1).

Biweekly therapy evolution in the COVID-19 first wave (1 March to 31 May) is shown in Table 2. A decrease in hospital stay, for both the hospitalization period (14 (5–32) vs. 7 (4–10) days) and length of stay in the ICU (19 (7–31) vs. 4 (1–12) days), was observed during the entire research period. Decreases in pneumonia (34.19% vs. 22.75%) and death (69.23% vs. 15.08%) were also observed in the same period (Figure 1).

### 2.2. Pharmacological Treatment

Antibiotics were used by 90.83% of the patients, 69.74% antimalarials, 44.37% steroids, 42.63% lopinavir-ritonavir, 9.37% tocilizumab and 7.34% anti-SIRS drugs. With the exception of antibiotics, the use of these medicines was higher in men than in women (Table 1).

The most commonly used antibiotics were ceftriaxone (69.28%), azithromycin (67.93%) and levofloxacin (15.2%). Among the antimalarials, hydroxychloroquine was used 10 times higher than chloroquine (64.95% vs. 6.13%). Regarding steroids, methylprednisolone (41.22%) was the most used, followed by prednisone (9.36%). With respect to the anti-SIRS drugs, IFN β-1b represented 5.64% of the total use (Table 3).

These medicines were used more commonly to treat interstitial pneumonia, except for steroids, with a marked difference observed in the use of tocilizumab (13.57% vs. 7.89%, *p* = 0.001) (Table 4). On the other hand, antibiotics, antimalarials, lopinavir-ritonavir and tocilizumab were used more commonly in non-deceased patients, compared to steroids and anti-SIRS drugs in deceased individuals (Table 4).

Regarding the evolution of medicines prescribed through the research period (Table 2), antibiotic use remained stable, with a small increase between 15 and 31 March, coinciding with a peak use of antimalarials, lopinavir-ritonavir and tocilizumab. Meanwhile, steroid use remained constant, with a maximum peak of use between 1 and 15 March. The use of anti-SIRS drugs, in particular IFN β-1b, decreased from 15.38% in early March to 0% in May.

### 2.3. Risk Factor for Clinical Outcomes and Medication Prescribed

Among the COVID-19 patients, death was more likely to occur in those over 65 years of age (OR: 9.05), in males (OR: 1.18), in those needing ventilation (OR: 3.59), and in those treated with anti-SIRS drugs (OR: 2.35), steroids (OR: 1.5), and tocilizumab (OR: 1.34).

On the other hand, for those 65 years of age or older (OR: 1.58), the need for ventilation (OR: 5.14), tocilizumab use (OR: 1.97), and steroids use (OR: 1.56) were risk factors for patients with SARS. In addition, AKI was more likely found among COVID-19 patients of 65 years of age or older (OR: 4.54), with mechanical ventilation (OR: 2.31) and the use of tocilizumab (OR: 1.55). Cardiomyopathy was more likely in the male gender (OR: 3.13) and in patients 65 years of age or older (OR: 1.8). Lastly, ventilation (OR: 2.77) and age (OR: 2.35) were risk factors for patients with shock (Figure 2).

## 3. Discussion

During the first COVID-19 wave, with a total population of 7307 patients, all groups of medicines were used according to the Spanish national guidelines [17,18]. 

Regarding the evolution of the medicines prescribed through this research period (Table 2), antibiotic and steroid use remained stable, while antimalarials, antivirals, tocilizumab and anti-SIRS drug use decreased.

Being 65 years of age or older, the need for ventilation and the use of tocilizumab and steroids are important risk factors for death and other adverse clinical outcomes among individuals in which SARS and AKI predominated.

This clinical picture produced by the analysis of real-world data information should be transmitted to clinicians in order to elaborate a consolidated response to mitigate the problems mentioned here, particularly considering that COVID-19 is a disease that is not still contained. More comprehensive assessments may be necessary to understand COVID-19 patients in our regions, and to establish differences with patients in other regions.

As in other COVID-19 patient cohorts, two from China [5,15], one from United Kingdom [31], one from the United States [32] and two national cohorts [33,34], the male gender was predominant. Nevertheless, the median age was 5 years higher in our region compared to the age among patients in Spain [33,34] or in United Kingdom [31], and 20 years higher than that in China [5,15]. There were also differences in the age of COVID-19 patients in Castile and Leon with respect to other patients in our country and Europe that may be due to the predominance of the elderly in our region.

Hospitalization period and length of stay in the ICU were consistent with other national results [34] and were similar to the United States [10], but, surprisingly, were small compared to overall COVID-19 patients in Spain [33]. In our hospitals, the provision of additional ICU beds (but not accounted for in EHRs) may also explain the low ratio of ICU patients observed [34].

Frequency of mortality (24.43%) was similar compared to other Spanish COVID-19 patient populations [33,34], the United Kingdom [31] and the United States [32]. However, mortality was higher than that reported for other European populations [35] and in Chinese [5,15] cohorts, and for all cases this was significantly higher in men than in women.

Use of all types of mechanical ventilation, such as oxygenation, NIPPV or IMV, were slightly lower than in other national cohorts [33,34] and elsewhere [5]. As in other reports, SARS and AKI were frequent [31,32,33,34,36,37].

Pharmacological treatment in hospitalized COVID-19 patients in the course of the first wave was based primarily on the disease symptoms, particularly addressing the prevention of respiratory failure [4]. Medicines used were and are currently authorized for other indications, but new biomolecules were also tried. Protocols for using medicines followed scientific evidence, and modifications were performed according to drug availability [18].

As in other areas of Spain [33,34], most patients (90.83%) received antibiotics, especially ceftriaxone, azithromycin, and levofloxacin, mostly indicated in severe cases to prevent bacterial superinfection [17]. However, no evidence exists to date on the prophylaxis effects for avoiding bacterial superinfection, and high mortality rates were reported in patients with secondary infections [38].

A low use of lopinavir-ritonavir compared to that described in Spain were noted [14,33,34,39]. In clinical trials, lopinavir-ritonavir were not associated with clinical improvement or decreased mortality in COVID-19 critically ill patients [40]. However, it seems that the triple antiviral therapy, lopinavir-ritonavir + IFN β-1b + ribavirin, is superior to lopinavir-ritonavir alone, considered for relieving of symptoms and thus facilitating the discharge of patients with mild to moderate COVID-19 [41].

With respect to remdesevir, which inhibits the replication of multiple coronaviruses in respiratory epithelial cells [26], since its development to treat Ebola [16], this medicine has shown a recovery time reduction in critically ill patients compared to those receiving the placebo [42,43,44,45]. Notwithstanding, remdesevir was used less in our region and with compassionate intention [18].

Most importantly, COVID-19 induces a cytokine storm and lymphopenia [46], with elevation of cytokine levels such as the IL-2 receptor, IL-6, IL-8, IL-10 and necrosis factor tumor-alpha (TNF α) [27], explaining SARS and multi-organ failure [28,39]. These immunological alterations can be treated by immunosuppressive and immunomodulatory drugs, such as antimalarials; steroids; IL-6R antagonists, such as tocilizumab; IL-1 antagonists, such as anakinra; and JAK inhibitors, such as ruxolitinib or baricitinib [47]. According to our results, two thirds of the patients were treated with antimalarials, which is consistent with the data observed in other investigations [33,34]. Indeed, chloroquine limits the replication of SARS-CoV-2 [48,49], with clear in vitro effects from hydroxychloroquine [50]. However, data on its clinical efficacy are contradictory [51,52], and in some cases its use is associated with an increase in the period of hospitalization and a high ventilation risk [53].

Half of the COVID-19 patients were treated with the combination hydroxychloroquine + azithromycin, which is associated with a significant decrease in mortality [24]. In these cases, regular evaluation of the electrocardiogram is mandatory, due to a possible increase in the QT interval, which is extensive in the use of fluoroquinolones [4,23,24].

Steroids were used in 44.37% of the observed patients, mainly methylprednisolone, with a higher consumption in those who died, and this finding is consistent with overall national data [33,34]. Steroids are frequently used as an adjuvant treatment for viral pneumonia [54], although steroids were associated with increased mortality, secondary infections and other complications in SARS-CoV and MERS-CoV [4]. Steroids favor inflammation control in COVID-19 patients [14]. The RECOVERY trial results indicate that steroids reduce the risk of death, artificial ventilation and a long period of hospitalization [22], and a meta-analysis found that the steroids are associated with lower 28-day all-cause mortality [55]. In any case, according to international guidelines, steroids should be started at low doses and used for reduced time periods [4,47].

Another medicine used was tocilizumab, which is authorized for rheumatoid arthritis and the cytokine release syndrome associated with CAR-T [47]. This medicine plays an important role in the cytokine storm caused by COVID-19 [56]. Although current evidence is scarce, tocilizumab is recommended to stop the inflammatory symptoms, and to reduce the need for artificial ventilation [18,57]. Consumption data in our region are consistent with the overall national data [33,34].

The use of other anti-SIRS drugs was much lower, highlighting only the use of IFN β-1b, with 5.64%. This medicine was used mainly at the beginning of the pandemic between 1 March and 31 March, considering its use against hepatitis C and its potential adverse effects [58]. Consumption decreased until discontinuation since there is in vitro evidence that IFN can increase the expression of the receptor for angiotensin converting enzyme 2 (ACE2) in human epithelial cells and favor infection [59].

Regarding the rest of anti-SIRS drugs, minimal use of anakinra, baricitinib, siltuximab and ruxolitinib was observed, probably related to the lack of solid clinical evidence.

As depicted by this study, the largest number of patients throughout the first wave of the pandemic was concentrated between 15 March and 30 April, with an accrual of severe cases treated with medicines.

Regarding our multivariate analysis, as in other studies, the male gender is a risk factor for death and cardiomyopathy variables, being logical in the latter case due to a higher prevalence of cardiovascular diseases in males [5,31,32,33,34].

In all clinical outcomes, death was more likely to occur in ages over 65 years, with an especially high OR in the death variable [31,33,34]. Obviously, the need for artificial ventilation was a risk factor for all clinical outcomes except cardiomyopathy, since it is associated with more severe patients and with a worse prognosis and disease evolution.

Regarding the results obtained for the different medicines used, only antimalarials and the combination of lopinavir-ritonavir presented an OR < 1, while tocilizumab, steroids and anti-SIRS drugs were associated with certain clinical outcomes.

Lastly, the limitations of any observational study must be mentioned. Registries and other real-world data sources should be considered as “emerging sources”, especially affected by confounders [60]. First, there may be selection bias, since for unknown reasons the medication record of some hospitalized COVID-19 patients was not collected. On the other hand, medicine selection was carried out in accordance with the Spanish guidelines [17,18], but, in daily clinical practice, other drugs could have been used for the pharmacological treatment of COVID-19, and were not included in this study.

## 4. Materials and Methods

### 4.1. Real-World Study Details

Using a real-world data study design, an epidemiological population-based registry study was carried out and findings are presented here in strict accordance to the Reporting of Studies Conducted using Observational Routinely Collected Data (RECORD) recommendations [61].

The research period covered the first COVID-19 wave in Spain, which occurred between 1 March and 31 May 2020. In this period, all the admitted COVID-19 patients in Castile and Leon public hospitals were considered. In this sense, to observe the evolution of the findings, biweekly periods were established.

We had access to JIMENA, the Castile and Leon electronic history record system, the Basic Minimum Data Set of Hospital Discharges registry (https://pestadistico.inteligenciadegestion.mscbs.es/publicoSNS/N/rae-cmbd/rae-cmbd) and to the CONCYLIA pharmaceutical care information system (http://www.saludcastillayleon.es/portalmedicamento/es/indicadoresinformes/concylia), which contains data on hospital medicines dispensing.

### 4.2. Variables

The age and gender of the patients were recorded. Different hospital stay variables were recorded, such as the hospitalization period (in days) and length of stay in ICU (in days). In addition, ventilation mode data was obtained for both oxygenation groups, NIPPV and IMV. The following clinical outcomes were considered: SARS, AKI, cardiomyopathy, shock, bacterial and fungal superinfection, disseminated intravascular coagulation (DIC) and death.

Information on the medicines was based on the Spanish guidelines [17,18] and treatment protocols from the Castile and Leon public hospitals (Appendix A). Hospital dispensing data were assessed according to the Anatomical Therapeutic Chemical code (ATC); the medicines were then grouped in categories for a better understanding of the findings (Table 5), namely, (1) antibiotics; (2) antimalarials; (3) steroids; (4) antivirals; (5) tocilizumab; and (6) anti-SIRS drugs. The rationale for separating tocilizumab from other anti-SIRS drugs it is that this medication was determined as the first preferred medication, and generally the most used when available.

Approval by our local ethics committee on 11 June 2020 (reference number PI 20-1863) was obtained.

### 4.3. Statistical Analysis

All analyses were done considering the age and gender distributions. In relation to patient age, two groups with a cut-off age of 65 years were defined.

Results are expressed as frequencies (percentage) with their corresponding 95% confidence interval (95% CI), as means accompanied by their standard deviations (SD) or as medians accompanied by their interquartile range (IQR) in the cases of asymmetrical distributions of the variable data (Kolmogorov–Smirnov and Shapiro–Wilk tests were used)

Differences between continuous variables were calculated using Student’s *t*-Test or the Mann–Whitney U test, and those between categorical variables by Pearson’s Chi-squared test or Fisher’s exact test, as appropriate.

A multivariate logistic regression model was performed using the forward method to identify predictors for the defined clinical outcomes dead, SARS, acute kidney injury (AKI), cardiomyopathy and shock. In the model, the variables of age (>65), gender, the need for ventilation and the variables related to medication use (antibiotics, antimalarials, steroids, antivirals (lopinavir-ritonavir), tocilizumab and anti-SIRS drugs) were included in the model. The results are shown in terms of Odds Ratio (OR) and 95% CI.

The level of significance was set at *p* ≤ 0.05. All statistical analyses were performed by using the Statistical Package for the Social Sciences (SPSS version 24.0., SPSS Inc, Chicago, IL, USA). Lastly, Microsoft Word and Excel (Microsoft Office version 365, Microsoft, Redmon, WA, USA) were used for preparing this manuscript.

## 5. Conclusions

The “health system collapse” caused by the COVID-19 pandemic in Spain forced rapid therapeutic decisions and pharmacological protocol development based on poor available scientific evidence. These protocols have been revised frequently based on the clinical experience acquired and the availability of specific drugs. In this way, the patterns of use of certain groups of medication changed from the beginning to the end of the COVID-19 first wave, with a decrease in the use of all drug groups also observed, with the exception of antibiotics and steroids, which remained constant.

In this research, information about the medicines used in the Castile and Leon hospital network was collected as a starting point to unify the criteria of the protocols established in each hospital, and to be better prepare for the next waves of COVID-19.

The information from the first wave shows that, in general, age and gender, as well as the need for ventilation, were related to the worst evolution and prognosis of the disease. Those that were prescribed anti-SIRS drugs, including tocilizumab, were more likely to suffer various disorders, as well as death (in this case also treated with steroids).

## Figures and Tables

**Figure 1 ijerph-17-09360-f001:**
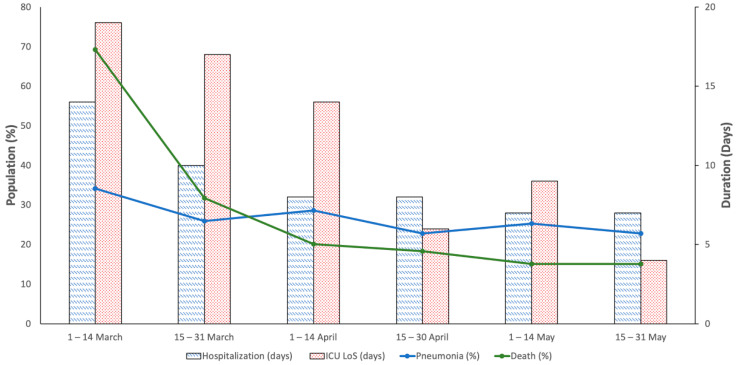
Biweekly clinical baselines in hospitalized COVID-19 patients in Castile and Leon (Spain) during the first wave (1 March to 31 May 2020).

**Figure 2 ijerph-17-09360-f002:**
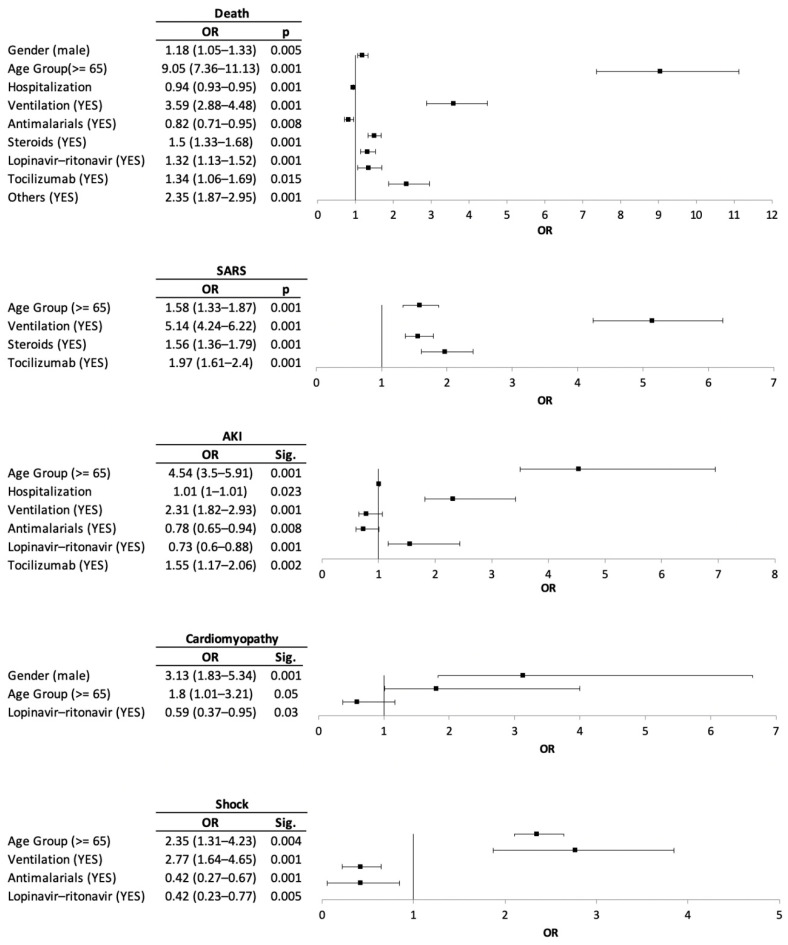
Risk factors for the clinical outcomes and medication prescribed.

**Table 1 ijerph-17-09360-t001:** Baseline characteristics and clinical outcomes of the hospitalized COVID-19 patients in Castile and Leon (Spain) during the first wave (1 March to 31 May 2020).

	TOTAL	MALE	FEMALE	*p*
***n***	7307	4169	3138	
Age (median and IQR)	76 (63–86)	75 (63–84)	79 (64–88)	0.001
Age < 65 (95% CI)	27.23 (26.21–28.25)	27.87 (26.51–29.23)	26.39 (24.84–27.93)	0.158
Age >= 65 (95% CI)	72.77 (71.75–73.79)	72.13 (70.77–73.49)	73.61 (72.07–75.16)	0.158
**Treatment**				
Oxygenation and ventilation (95% CI)				
Oxygenation only	2.52 (2.16–2.88)	2.85 (2.35–3.36)	2.07 (1.57–2.57)	0.034
NIPPV	1.63 (1.34–1.92)	2.16 (1.72–2.6)	0.92 (0.59–1.26)	0.001
IMV	3.5 (3.08–3.93)	4.73 (4.08–5.37)	1.88 (1.4–2.36)	0.001
*Drugs (95% CI)*				
Antibiotics	90.83 (90.17–91.49)	90.19 (89.29–91.09)	91.68 (90.72–92.65)	0.029
Antimalarial	69.74 (68.69–70.79)	71.7 (70.33–73.06)	67.14 (65.5–68.79)	0.001
Steroids	44.37 (43.23–45.51)	47.83 (46.31–49.35)	39.77 (38.06–41.48)	0.001
Antivirals	42.63 (41.52–43.93)	45.42 (43.5–46.93)	38.95 (37.22–40.66)	0.001
Tocilizumab	9.37 (8.71–10.04)	12.14 (11.15–13.13)	5.7 (4.89–6.52)	0.001
Others anti SIRS *	7.34 (6.74–7.93)	9.35 (8.47–10.24)	4.65 (3.92–5.39)	0.001
**Outcomes**				
Hospitalization days (median and IQR)	9 (5–15)	9 (5–15)	8 (5–14)	0.001
ICU LoS (median and IQR)	15 (7–30)	15 (7–32)	15 (8–24)	0.359
SARS (95% CI)	14.03 (13.23–14.82)	15.59 (14.49–16.69)	11.95 (10.82–13.09)	0.001
AKI (95% CI)	10.87 (10.15–11.58)	11.3 (10.34–12.26)	10.29 (9.23–11.36)	0.172
Cardiomyopathy (95% CI)	1.15 (0.91–1.39)	1.61 (1.23–1.99)	0.54 (0.28–0.8)	0.001
Shock (95% CI)	1.51 (1.23–1.78)	1.66 (1.27–2.04)	1.31 (0.91–1.7)	0.226
Bacterial superinfection (95% CI)	3.59 (3.16–4.01)	3.89 (3.3–4.47)	3.19 (2.57–3.8)	0.112
Fungal superinfection (95% CI)	2.23 (1.89–2.57)	2.11 (1.67–2.55)	2.39 (1.86–2.92)	0.424
DIC (95% CI)	0.18 (0.08–0.27)	0.29 (0.13–0.45)	0.03 (0.01–0.05)	0.001
Death (95% CI)	24.43 (23.44–25.41)	26.12 (24.79–27.45)	22.18 (20.73–23.63)	0.001

* Anakinra, baricitinib, interferon, ruxolitinib, siltuximab. Abbreviations: 95% CI, confidence interval; IQR, interquartile range; SIRS, systemic inflammatory response syndrome; NIPPV, noninvasive positive pressure ventilation; IMV, invasive mechanical ventilation; ICU, intensive care unit; LoS, length of stay; SARS, severe acute respiratory syndrome; AKI, acute kidney injury; DIC, disseminated intravascular coagulation.

**Table 2 ijerph-17-09360-t002:** Biweekly therapy evolution in the hospitalized COVID-19 patients in Castile and Leon (Spain) during the first wave (1 March to 31 May 2020).

Medicines	1–14 March	15–31 March	1–14 April	15–30 April	1–14 May	15–31 May
	*n* = 117	*n* = 2819	*n* = 2155	*n* = 1250	*n* = 588	*n* = 378
Antibiotics	79.49 (72.17–86.8)	91.63 (90.61–92.65)	90.95 (89.74–92.16)	91.6 (90.06–93.14)	88.1 (85.48–90.71)	89.42 (86.32–92.52)
Ceftriaxone	47.01 (37.96–56.05)	71.83 (70.17–73.49)	68.45 (66.48–70.41)	66.96 (64.35–69.57)	69.73 (66.01–73.44)	68.78 (64.11–73.45)
Azithromycin	25.64 (17.73–33.55)	70.95 (69.27–72.62)	73.74 (71.88–75.59)	67.28 (64.68–69.88)	52.21 (48.17–56.25)	52.12 (47.08–57.15)
Levofloxacin	44.44 (35.44–53.45)	16.42 (15.06–17.79)	10.95 (9.63–12.27)	15.52 (13.51–17.53)	17.52 (14.44–20.59)	17.72 (13.88–21.57)
Cefditoren	0.85 (0.1–1.69)	3.12 (2.48–3.76)	2.55 (1.89–3.22)	2.4 (1.55–3.25)	2.04 (0.9–3.18)	1.32 (0.17–2.47)
Teicoplanin	2.56 (0.3–5.49)	1.21 (0.8–1.61)	1.62 (1.09–2.16)	1.36 (0.72–2)	1.19 (0.31–2.07)	1.59 (0.33–2.85)
Clarithromycin	0.85 (0.1–1.69)	0.32 (0.11–0.53)	0.19 (0–0.37)	0.48 (0.1–0.86)	0.34 (0.13–0.55)	1.32 (0.17–2.47)
Cefotaxime	1.71 (0.03–2.65)	0.14 (0–0.28)	0.23 (0.03–0.44)	0.4 (0.05–0.75)	0.51 (0.07–0.95)	0.26 (0.05–0.58)
Moxifloxacin	0 (0–0)	0.18 (0.02–0.33)	0.42 (0.15–0.69)	0.24 (0.03–0.45)	0.51 (0.07–0.95)	0.26 (0.25–0.58)
Ceftaroline	0 (0–0)	0.11 (0.01–0.21)	0 (0–0)	0.08 (0.02–0.14)	0 (0–0)	0 (0–0)
Antimalarials	42.74 (33.77–51.7)	84.04 (82.68–85.39)	76.29 (74.49–78.08)	60 (57.28–62.72)	32.14 (28.37–35.92)	24.87 (20.51–29.23)
Hydroxycloroquine	36.75 (28.02–45.49)	76.48 (74.92–78.05)	71.18 (69.27–73.1)	58.96 (56.23–61.69)	31.63 (27.87–35.39)	23.81 (19.52–28.1)
Cloroquine	6.84 (2.26–11.41)	10.85 (9.71–12)	5.15 (4.22–6.08)	1.12 (0.54–1.7)	0.51 (0.07–0.95)	1.59 (0.33–2.85)
Steroids	59.83 (50.95–68.71)	44.45 (42.61–46.28)	43.67 (41.57–45.76)	44.72 (41.96–47.48)	42.69 (38.69–46.68)	44.44 (39.44–49.45)
Methylprednisolone	56.41 (47.43–65.4)	42.21 (40.39–44.04)	40.23 (38.16–42.3)	41.52 (38.79–44.25)	37.59 (33.67–41.5)	39.42 (34.49–44.34)
Prednisone	11.11 (5.42–16.81)	8.09 (7.08–9.09)	10.12 (8.84–11.39)	9.44 (7.82–11.06)	11.56 (8.98–14.15)	10.32 (7.25–13.38)
Antivirals	38.46 (29.65–47.28)	65.02 (63.01–66.43)	42.51 (40.25–44.32)	20.24 (18.01–22.47)	8.5 (6.25–10.76)	6.61 (4.11–9.12)
Lopinavir-Ritonavir	38.46 (29.65–47.28)	64.88 (63.12–66.64)	42.37 (40.28–44.45)	20.24 (18.01–22.47)	8.5 (6.25–10.76)	6.61 (4.11–9.12)
Remdesevir	0 (0–0)	0.14 (0–0.28)	0.14 (0.02–0.26)	0 (0–0)	0 (0–0)	0 (0–0)
Tocilizumab	7.69 (2.86–12.52)	13.2 (11.95–14.45)	10.58 (9.28–11.88)	4.8 (3.61–5.99)	1.87 (0.78–2.97)	1.32 (0.17–2.47)
Other Anti-SIRS	15.38 (8.85–21.92)	13.2 (11.95–14.45)	5.06 (4.13–5.98)	2.16 (1.35–2.97)	1.02 (0.21–1.83)	1.06 (0.03–2.09)
Interferon Beta	15.38 (8.85–21.92)	12.2 (10.99–13.41)	2.18 (1.56–2.8)	0 (0–0)	0 (0–0)	0.79 (0.1–1.67)
Anakinra	0 (0–0)	0.89 (0.54–1.23)	2.51 (1.85–3.17)	1.68 (0.97–2.39)	0.17 (0.06–0.35)	0.53 (0.2–1.22)
Baricitinib	0 (0–0)	0.11 (0.01–0.21)	0.42 (0.15–0.69)	0.64 (0.2–1.08)	0.85 (0.11–1.59)	0 (0–0)
Siltuximab	0 (0–0)	0.18 (0.02–0.33)	0.23 (0.03–0.44)	0 (0–0)	0 (0–0)	0 (0–0)
Ruxolitinib	0 (0–0)	0.11 (0.01–0.21)	0.19 (0–0.37)	0.08 (0.02–0.14)	0 (0–0)	0 (0–0)

Abbreviations: 95% CI, confidence interval; SIRS, systemic inflammatory response syndrome.

**Table 3 ijerph-17-09360-t003:** Pharmacological treatment in hospitalized COVID-19 patients in Castile and Leon (Spain) during the first wave (1 March to 31 May 2020).

Medicines	TOTAL (95 CI)	No Death (95 CI)	Death (95 CI)	*p*
	*n* = 7307	*n* = 5522	*n* = 1785	
Antibiotics	90.83 (90.17–91.49)	91.22 (90.47–91.96)	89.64 (88.22–91.05)	0.044
Ceftriaxone	69.28 (68.22–70.33)	69.21 (68–70.43)	69.47 (67.33–71.6)	0.84
Azithromycin	67.93 (66.86–69)	69.68 (68.47–70.9)	62.52 (60.28–64.77)	0.001
Levofloxacin	15.26 (14.43–16.08)	13.94 (13.03–14.86)	19.33 (17.5–21.16)	0.001
Cefditoren	2.61 (2.25–2.98)	2.95 (2.51–3.4)	1.57 (0.99–2.15)	0.001
Teicoplanin	1.4 (1.13–1.66)	1.18 (0.89–1.46)	2.07 (1.41–2.73)	0.005
Clarithromycin	0.37 (0.23–0.51)	0.31 (0.16–0.45)	0.56 (0.21–0.91)	0.127
Moxifloxacin	0.29 (0.16–0.41)	0.29 (0.15–0.43)	0.28 (0.03–0.53)	0.947
Cefotaxime	0.27 (0.15–0.39)	0.33 (0.18–0.48)	0.11 (0.04–0.19)	0.133
Ceftaroline	0.05 (0–0.11)	0.07 (0–0.14)	0 (0–0)	
Antimalarials	69.74 (68.69–70.79)	71.01 (69.81–72.2)	65.83 (63.63–68.03)	0.001
Hydroxycloroquine	64.95 (63.86–66.05)	66.37 (65.12–67.62)	60.56 (58.29–62.83)	0.001
Cloroquine	6.13 (5.58–6.68)	5.94 (5.32–6.56)	6.72 (5.56–7.88)	0.231
Steroids	44.37 (43.23–45.51)	42.12 (40.82–43.42)	51.32 (49–53.64)	0.001
Methylprednisolone	41.22 (40.09–42.35)	38.59 (37.31–39.88)	49.36 (47.04–51.68)	0.001
Prednisone	9.36 (8.69–10.03)	10.45 (9.64–11.26)	5.99 (4.89–7.1)	0.001
Antivirals	42.64 (41.52–43.93)	42.99 (41.61–44.36)	41.91 (39.46–44.35)	0.35
Lopinavir-Ritonavir	42.63 (41.5–43.76)	42.92 (41.61–44.22)	41.74 (39.45–44.02)	0.38
Remdesevir	0.1 (0.02–0.17)	0.07 (0–0.14)	0.17 (0.01–0.33)	0.256
Tocilizumab	9.37 (8.71–10.04)	9.69 (8.91–10.47)	8.4 (7.12–9.69)	0.105
Other Anti-SIRS	7.34 (6.74–7.93)	6.25 (5.61–6.89)	10.7 (9.27–12.13)	0.001
Interfereon Beta	5.64 (5.11–6.17)	4.85 (4.29–5.42)	8.07 (6.8–9.33)	0.001
Anakinra	1.41 (1.14–1.68)	1.16 (0.88–1.44)	2.18 (1.51–2.86)	0.001
Baricitinib	0.34 (0.21–0.48)	0.33 (0.18–0.48)	0.39 (0.1–0.68)	0.677
Siltuximab	0.14 (0.05–0.22)	0.07 (0–0.14)	0.34 (0.07–0.6)	0.009
Ruxolitinib	0.11 (0.03–0.19)	0.07 (0–0.14)	0.22 (0–0.44)	0.092
Remdesevir	0.1 (0.02–0.17)	0.07 (0–0.14)	0.17 (0.01–0.33)	0.256

Abbreviations: 95% CI, confidence interval; SIRS, systemic inflammatory response syndrome.

**Table 4 ijerph-17-09360-t004:** Pharmacological treatment in hospitalized COVID-19 patients, with and without pneumonia, in Castile and Leon (Spain) during the first wave (1 March to 31 May 2020).

Drugs	No Pneumonia (95% CI)	Pneumonia (95% CI)	*p*
	*n* = 5399	*n* = 1908	
Antibiotics	90.31 (89.52–91.1)	92.3 (91.1–93.49)	0.001
Ceftriaxone	69.05 (67.82–70.28)	69.92 (67.86–71.97)	0.481
Azithromycin	66.64 (65.38–67.9)	71.59 (69.57–73.62)	0.001
Levofloxacin	15.1 (14.14–16.05)	15.72 (14.09–17.36)	0.512
Cefditoren	2.43 (2.02–2.84)	3.14 (2.36–3.93)	0.091
Teicoplanin	1.28 (0.98–1.58)	1.73 (1.14–2.31)	0.148
Clarithromycin	0.37 (0.21–0.53)	0.37 (0.1–0.64)	0.982
Moxifloxacin	0.28 (0.14–0.42)	0.31 (0.06–0.57)	0.797
Cefotaxime	0.28 (0.14–0.42)	0.26 (0.03–0.49)	0.91
Ceftaroline	0.06 (0.01–0.10)	0.05 (0.01–0.12)	0.96
Antimalarials	68.4 (67.16–69.64)	73.53 (71.55–75.51)	0.001
Hydroxycloroquine	63.12 (61.84–64.41)	70.13 (68.07–72.18)	0.001
Cloroquine	6.82 (6.14–7.49)	4.19 (3.29–5.09)	0.001
Steroids	44.67 (43.35–46)	43.5 (41.28–45.73)	0.375
Methylprednisolone	41.71 (40.4–43.03)	39.83 (37.64–42.03)	0.152
Prednisone	8.8 (8.04–9.55)	10.95 (9.55–12.36)	0.005
Antivirals	41.08 (39.34–42.35)	47.38 (44.92–49.43)	0.001
Lopinavir-Ritonavir	41.01 (39.7–42.32)	47.22 (44.98–49.46)	0.001
Remdesevir	0.07 (0–0.15)	0.16 (0.001–0.31)	0.313
Tocilizumab	7.89 (7.17–8.61)	13.57 (12.04–15.11)	0.001
Other Anti-SIRS	7.08 (6.39–7.76)	8.07 (6.85–9.29)	0.152
Interfereon Beta	5.52 (4.91–6.13)	5.97 (4.91–7.04)	0.459
Anakinra	1.28 (0.98–1.58)	1.78 (1.19–2.38)	0.108
Baricitinib	0.31 (0.17–0.46)	0.42 (0.13–0.71)	0.502
Ruxolitinib	0.09 (0.01–0.17)	0.16 (0.01–0.31)	0.469
Siltuximab	0.09 (0.01–0.17)	0.26 (0.03–0.49)	0.085

Abbreviations: 95% CI, confidence interval; SIRS, systemic inflammatory response syndrome.

**Table 5 ijerph-17-09360-t005:** List of medicines used in the COVID-19 treatment according to Spanish guidelines [17,18].

Medicines Group	ATC Code	Medicine	Medicines Group	ATC Code	Medicine
Antibiotics	J01DD01	Cefotaxime	Anti-SIRS Drugs	L01XE18	Ruxolitinib
	J01DD04	Ceftriaxone		L03AB05	Interferon alpha 2b
	J01DD16	Cefditoren		L03AB08	Interferon beta 1b
	J01DI02	Ceftaroline		L04AA37	Baricitinib
	J01FA09	Clarithromycin		L04AC03	Anakinra
	J01FA10	Azithromycin		L04AC07	Tocilizumab
	J01MA12	Levofloxacin		L04AC11	Siltuximab
	J01MA14	Moxifloxacin		L04AC14	Sarilumab
	J01XA02	Teicoplanine	Antivirals	J05AR10	Lopinavir/Ritonavir
Antimalarials	P01BA01	Chloroquine		J05AX95 *	Remdesivir
	P01BA02	Hidroxychloroquine	Steroids	H02AB04	Methylprednisolone
				H02AB07	Prednisone

Abbreviations: SIRS, systemic inflammatory response syndrome. * Provisional ATC Code.

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
