# Peer review of "Clinical Profile, Treatment and Predictors during the First COVID-19 Wave: A Population-Based Registry Analysis from Castile and Leon Hospitals"

_ijerph, 2020, doi:10.3390/ijerph17249360_

Round 1

Reviewer 1 Report

in this paper, predictors in the COVID-19 hospitalized patients from March 1stto May 31th2020 is presented through a population-based registry study in spain. The research period covered the first COVID-19 wave in Spain, which occurred between March 1stand May 31th2020. In this period, all admitted COVID-19 patients in Castile and Leon public hospitals were considered. In this sense, to observe the evolution of the findings, biweekly periods have been established. The mortality rate was found 24.43% in castilla and leon sector hospitals, according to data analysed, including anticovid medicines, the use of tocilizumab (9.37%) and anti-SIRS (systemic inflammatory response syndrome) medicines (7.34%) which were remarkable fighting against pandemic disease. In this study 7,307 hospitalized COVID-19 patients were included, of which 57.05% were men and 42.95% women (p=0.001). Pharmacological treatment and risky factor such as steroids use, age and pharmacotherapy were investigated and tables were constructed covering bi-weekly therapy evolution in the hospitalized COVID-19 patients in Castile and Leon (Spain) first wave, and Pharmacological treatment in hospitalized COVID-19 patients, with and without pneumonia, (March 1st-May 31th2020), Using a real-world data study design, this epidemiological population-based registry study was carried out and their findings are presented covering all in strict accordance to the Reporting of studies Conducted using Observational Routinely-collected Data (RECORD) recommendations. strenght:  Paper is sound and complete covering pandemia in a big part of Spain.weakness.  I reccomend to revise English language carefully and maybe to cover more people in Spain.

Reviewer 2 Report

Please see my comments in the attached pdf file.
